# Lifestyle and Metabolic Syndrome: Contribution of the Endocannabinoidome

**DOI:** 10.3390/nu11081956

**Published:** 2019-08-20

**Authors:** Vincenzo Di Marzo, Cristoforo Silvestri

**Affiliations:** 1École de nutrition, Université Laval, Québec, QC G1V 0A6, Canada; 2Institut sur la nutrition et les aliments fonctionnels, Université Laval, Québec, QC G1V 0A6, Canada; 3Canada Excellence Research Chair on the Microbiome-Endocannabinoidome Axis in Metabolic Health, Université Laval, Québec, QC G1V 0A6, Canada; 4Centre de recherche de l’Institut universitaire de cardiologie et de pneumologie de Québec, Québec, QC G1V 4G5, Canada; 5Department de médecine, Université Laval, Québec, QC G1V 0A6, Canada; 6Institute of Biomolecular Chemistry, Consiglio Nazionale delle Ricerche, 80078 Pozzuoli, Italy

**Keywords:** endocannabinoids, endocannabinoidome, metabolic syndrome, microbiome

## Abstract

Lifestyle is a well-known environmental factor that plays a major role in facilitating the development of metabolic syndrome or eventually exacerbating its consequences. Various lifestyle factors, especially changes in dietary habits, extreme temperatures, unusual light–dark cycles, substance abuse, and other stressful factors, are also established modifiers of the endocannabinoid system and its extended version, the endocannabinoidome. The endocannabinoidome is a complex lipid signaling system composed of a plethora (>100) of fatty acid-derived mediators and their receptors and anabolic and catabolic enzymes (>50 proteins) which are deeply involved in the control of energy metabolism and its pathological deviations. A strong link between the endocannabinoidome and another major player in metabolism and dysmetabolism, the gut microbiome, is also emerging. Here, we review several examples of how lifestyle modifications (westernized diets, lack or presence of certain nutritional factors, physical exercise, and the use of cannabis) can modulate the propensity to develop metabolic syndrome by modifying the crosstalk between the endocannabinoidome and the gut microbiome and, hence, how lifestyle interventions can provide new therapies against cardiometabolic risk by ensuring correct functioning of both these systems.

## 1. Introduction

Diets poor in essential nutritional factors (e.g., dietary fibers or vitamins) and rich in high-calorie nutrients, lack of exercise, and uncontrolled use of recreational substances or certain therapeutic drugs, together with other environmental challenges such as recently changed lifestyle habits in populations living at extreme temperatures or regarding night–day cycles, are all known to negatively affect the body’s ability to regulate energy metabolism and, hence, contribute to the development of metabolic syndrome [1]. A plethora of epidemiological studies point to these aspects as major predictors of various forms of dysmetabolism, including obesity and visceral adipose tissue accumulation [2], glucose intolerance, pre-diabetes and type 2 diabetes [3], dyslipidemia [4], hypertension [5] and, eventually, the development of atherogenic inflammation [6] and the ensuing cardiovascular disorders [7]. By contrast, several other studies show how fighting bad dietary habits and the introduction of some dietary supplements and vitamins, as well as the increase of physical exercise, can successfully counteract many features of metabolic syndrome [1,8,9].

At the same time, multifaceted lifestyle aspects are emerging as having a strong impact on an endogenous system of lipid signals known as the endocannabinoid system and its more recent expansion to the endocannabinoidome (see below), which play an important role in several physiological and pathological conditions and, particularly, in the control of energy metabolism and its dysfunctions [10,11]. Endocannabinoids and endocannabinoidome mediators are ultimately derived from long-chain fatty acids, and it is therefore predictable that prolonged diets rich in some fatty acids rather than others can affect the tissue concentrations of these molecules in as much as they can change the fatty acid composition of phospholipids acting as biosynthetic precursors [12,13]. Additionally, there is evidence that pre- and probiotics can produce beneficial effects partly mediated by endocannabinoidome mediators, pointing to the possibility that at least some of the numerous physiological and pathological actions respectively displayed by a healthy or disrupted gut microbiota (known as dysbiosis) may be due to changes in this complex system of lipid chemical signals, both at the central nervous system and peripheral tissue level. This seems to be particularly true in the context of metabolic control in which the intestinal flora, like the endocannabinoidome, is known to play a major role [14,15,16]. This evidence is reinforced by the recent finding that some commensal bacteria produce endocannabinoid-like compounds able to activate the same receptors as their host cell counterparts [17]. Conversely, in mice, pharmacological or tissue-selective genetic manipulation of the tissue concentrations and receptor-mediated activity of endocannabinoids and endocannabinoid-like molecules was found to affect, at the same time, the relative composition in phyla, orders, genera, and species of microorganisms that populate the intestinal tract as well as the metabolic response to high-fat diets [18,19,20,21]. If one considers that gut microbiota composition is altered by the same dietary and environmental factors and unhealthy behaviors that affect the endocannabinoid system [20,22,23,24], then it is perhaps not so farfetched to suggest that the lifestyle–gut microbiome–endocannabinoidome triangle plays a crucial role in the development of metabolic syndrome.

In this article, we shall discuss several ways through which lifestyle-induced alterations of the endocannabinoidome—very often through direct or indirect effects on the gut microbiome (µB; that is the ensemble of genes, proteins, and metabolites provided by intestinal microorganisms)—can either worsen or ameliorate energy metabolism in mammals and, hence, influence the development of the metabolic syndrome.

## 2. The Endocannabinoidome

The very popular drug of abuse, marijuana, is prepared from the flowers of *Cannabis sativa* varieties containing relatively high contents of the non-psychotropic precursor of Δ^9^-tetrahydrocannabinol (THC), i.e., Δ^9^-tetrahydrocannabinolic acid, wherefrom the better known THC is obtained following desiccation and/or heating. However, *Cannabis sativa*—including those varieties that have been used for centuries for their fibers and employed to make ropes and paper—contains more than one hundred other THC and THC acid-like compounds in the inflorescence. These compounds have little or no psychotropic action and, together with THC and THC-acid, are known as *cannabinoids*. The euphoric, appetite-stimulating, and many other “central” actions of THC, are due to its unique capability to bind and activate a G-protein-coupled receptor (GPCR), the cannabinoid receptor type-1 (CB1), whereas another GPCR, the cannabinoid receptor type-2 (CB2), with little more than 50% homology with CB1 [1,2], is responsible for the immune-modulatory effects of this compound. So far, THC is the only plant-derived cannabinoid known to be capable of potently and efficaciously activating these receptors (which is why they should, in our opinion, be renamed “THC receptors”), although a THC congener, Δ^9^-tetrahydrocannabivarine (THCV), was more recently shown to antagonize CB1 [25]. The discovery of cannabinoid receptors suggested the existence of endogenous ligands for such receptors. Two small lipids ultimately derived from arachidonic acid, *N*-arachidonoylethanolamine (AEA or anandamide) and 2-arachidonoylglycerol (2-AG), were indeed identified and shown to be capable of high-affinity binding to both CB1 and CB2 receptors, stimulating their activity with good efficacy [26,27]. These molecules were named *endocannabinoids* (eCBs) [28].

The eCBs come with their own anabolic and catabolic routes and enzymes, biosynthetic precursors, and hydrolysis products, which are inactive at cannabinoid receptors. By the turn of the last century, it was established that AEA is biosynthesized from the hydrolysis of *N*-arachidonoyl-phosphatidylethanolamines catalyzed by an *N*-acyl-phosphatidylethanolamine-specific phospholipase D-like enzyme (NAPE-PLD), whereas 2-AG is produced from the hydrolysis of 1-acyl-*sn*-2-arachidonoyl-glycerols (AcArGs), catalyzed by either *sn*-1 selective diacylglycerol lipase-α or -β (DAGLα or DAGLβ). AEA is hydrolyzed to arachidonic acid (AA) and ethanolamine by fatty acid amide hydrolase (FAAH), and 2-AG to AA and glycerol by monoacylglycerol lipase (MAGL) [29,30,31,32]. This ensemble of lipids, enzymes, and CB1 and CB2 receptors is known as the “endocannabinoid system”. While the enzymes mentioned above are historically considered to be the canonical ones that regulate endocannabinoid levels, it must be noted that other pathways have also been identified (see Figure 1B, recently reviewed in [33]). For example, AEA may be synthesized by the combined action of ABDH4 and GDE1 [34] or PTPN22 [35].

It was soon realized that AEA and 2-AG, like several other lipid mediators, are quite promiscuous in their pharmacological activity in as much as they were suggested to modulate the activity of other proteins at concentrations often, but not necessarily, higher than those required to activate CB1 and CB2. These receptors were later found to often be even better targets for some of the congeners of AEA and 2-AG, i.e., the long-chain *N*-acylethanolamines (NAEs) and 2-monoacylglycerols (2-MAGs), respectively, and include (1) thermosensitive transient receptor potential (TRP) channels, such as the “capsaicin receptor”, or TRP of vanilloid type-1 (TRPV1), the “menthol receptor”, or TRP of melastatin type-8 (TRPM8), and the TRP of vanilloid type-2 (TRPV2) channels, as well as the T-type Ca^2+^ channel (Ca_v.3.1_); (2) some orphan GPCRs, such as GPR55, GPR110, or GPR119; and (3) peroxisome proliferator-activated receptor-α and -γ (PPARα and PPARγ) (Figure 1A; recently reviewed in [33]. The eCB congeners, which are biosynthesized using NAPE-PLD or DAGLs from precursors similar to those of the two eCBs, and inactivated to the respective fatty acids and ethanolamine or glycerol by FAAH and MAGL, can also be produced and degraded via alternative pathways and enzymes, and, as mentioned above, this also applies to AEA and 2-AG (Figure 1B). Finally, several other long-chain fatty acid derivatives have also been identified during the last 15 years, including primary fatty acid amides and several *N*-acylated amino acids and neurotransmitters that often share molecular targets and/or inactivating enzymes with eCBs (Figure 1A,B). These findings led to the definition of the “expanded eCB system” or *endocannabinoidome* (eCBome), which includes a plethora of lipid mediators (including some enzymatic oxidation products of AEA and 2-AG) and tens of proteins acting as biosynthetic and inactivating enzymes, or molecular targets, for these mediators (recently reviewed in [33]).

The existence of the eCBome complicates the development of selective pharmacological and genetic tools to be used for the understanding of the several tissue-specific local functions of the eCBs, and for the exploitation of this knowledge for the development of new therapies against pathological conditions in which AEA and 2-AG are involved. On the other hand, if one looks at the eCBome as a whole and as the potential target of several physiopathological and environmental clues, and at eCBome profiles as possible personalized fingerprints of disease and responses to lifestyle, this complex signaling *hypersystem*, no matter how challenging, may open new therapeutic and diagnostic avenues. Indeed, as will be discussed below, diet and dietary components, habits, exercise, and the environment strongly impact on the eCBome—to an extent of which we have had perhaps, so far, only a partial view.

## 3. Dietary Fats and the Endocannabinoidome

In the obese state, the eCB system is modulated at the level of anabolic and catabolic enzyme activity, endocannabinoid levels, and CB1 receptor expression, resulting in a generally increased eCB tone “in the wrong place and at the wrong time” [36]. BMI positively correlates with circulating AEA and 2-AG levels, especially when fat distribution is partitioned more towards intra-abdominal stores [37,38,39]. However, the levels of AEA are dysregulated in obesity with respect to responses to feeding or the time of day as viscerally obese men were found to have significantly lower levels of AEA in the morning than normoweights [38]. The observed increases in AEA and 2-AG levels appear to be due to changes in expression of adipose tissue-metabolizing enzymes, as the AEA-catabolizing enzyme FAAH was decreased and the 2-AG-anabolizing enzyme DAGLα was increased in the adipose tissue from obese individuals in conjunction with decreased CB1 expression, perhaps as a homeostatic compensatory response [37,39,40]. Changes in eCBome gene expression within adipose tissue appear to be depot-specific, however, since gluteal subcutaneous adipose tissue from obese subjects had decreased eCBome gene expression (including FAAH, DAGLα, and CB1) while abdominal subcutaneous adipose tissue showed the opposite trend, with visceral adipose tissue similarly having increased CB1 expression [41].

Obesogenic diets characterized by high fat content are increasingly prevalent in westernized societies. High-fat diets increase AEA and/or 2-AG levels [12,13]. While *N*-oleoylethanolamine (OEA), *N*-palmitoylethanolamine (PEA), and *N*-linoleoylethanolamine (LEA) levels are reduced in the jejunum and/or stomach in response to 1 week [42] or up to 8 weeks [43] of high-fat feeding, prolonged feeding (14 weeks) increased OEA levels in the stomach concomitant with increased NAPE-PLD and decreased FAAH expression [43]. In the liver, a high-fat diet increased AEA levels and CB1 signaling, which contributed to the activation of genetic programs that increase fatty acid production [13]. In a very recent study in which circulating eCBome levels were tracked over time in mice on a high-fat diet, AEA, PEA, and *N*-docosahexanoylethanolamine (DHEA) levels increased rapidly over the course of a week, while SEA and 2-AG increases became significant only after 4 weeks and, finally, OEA increased after 10 weeks [44]. While gene expression changes in eCBome enzymes were observed in muscle and liver tissues, they were transient; however, the expression of the 2-AG anabolic enzyme DAGLβ was constantly increased in white and brown adipose tissue (BAT) from 4 weeks, while the NAE anabolic enzyme NAPE-PLD was constantly increased only in the BAT from 3 days on [44]. This study supports the conclusions inferred from human studies, that adipose tissue is one of the main regulators of circulating 2-AG in obesity. The potential contribution of BAT (at least in mice) to the regulation of NAEs was a surprising result, though at least in the case of OEA, it cannot be ruled out that the intestinal tract is one of the major sources [43].

Changes in eCBome mediator levels in response to high-fat feeding occur very quickly in mice. Recently, Everard et al. showed that after just 4 h of initial high-fat-diet feeding, jejunal AEA and 2-AG levels decreased while OEA, 2-OG, 2-LG, and 2-PG levels increased [18], but after 5 weeks of exposure AEA, LEA increased, as did 2-OG and 2-PG.

In utero or neonatal exposure to dietary perturbations can have long-lasting effects on an individual and, indeed, the eCBome is significantly impacted by exposure to high-fat diets early on in life with long-lasting consequences. Maternal high-fat feeding resulted in sustained elevation of CB1/2, FAAH, and MAGL levels in the livers of adult male rats, with changes in redox homeostasis [45]. Maternal exposure to high-fat diet also increased CB1 in the male, and CB2 in the female hypothalamus at birth, while CB1 and FAAH expression were increased, and CB2 and MGLL were decreased in the BAT of males and females, respectively. Both sexes developed an increased adiposity and preference for high-fat diets [46].

Dietary linoleic acid (LA) is a major *n*-6 fatty acid component of Western diets, making up over 80% of the polyunsaturated fatty acid (PUFA) consumed in the United States [47], resulting in an imbalance the ratio of *n*-6/*n*-3 fatty acids consumed greatly in favor of the former. LA is linked to obesity and is efficiently converted to the AEA and 2-AG constituent arachidonic acid (AA), thus explaining its ability to increase AEA and 2-AG levels and to produce obesogenic effects [48,49]. Indeed, even within the context of a low-fat diet, high levels of LA increased liver AEA and 2-AG levels, promoting obesity and associated adipose tissue inflammation [50]. Inclusion of *n*-3 fatty acids to an LA-rich diet reverses the latter’s effects on AEA and 2-AG levels [48]. Similar results have been obtained with EPA/DHA *n*-3 fatty acid-rich krill and, to a lesser extent, fish oil [12,51,52,53]. Additionally, supplementing young mice on a lard diet with flax seed oil rich in the *n*-3 fatty acid α-linolenic acid significantly decreased liver AEA levels and improved glucose homeostasis after a subsequent 10 weeks on a high-lard diet [54]. These effects are believed to largely be the result of decreasing the *n*-6/*n*-3 PUFA ratio, which results in AA displacement from phospholipid membranes, thus reducing the amounts of the biosynthetic precursors of AEA and 2-AG. In support of this, *n*-3 PUFAs provided as phospholipids, rather than free fatty acids, result in more significant decreases in eCB levels [12,52]. Correspondingly, decreasing *n*-3 PUFA phospholipid content increased 2-AG liver levels and promoted hepatosteatosis and insulin resistance [55].

These data suggest that some of the therapeutic properties against metabolic disorders (such as against high triglycerides) of *n*-3 fatty PUFA may be ascribed to a reduction of eCB overactivity, and this has also been suggested to be the case in obese humans [53]. However, the metabolic benefits of dietary *n*-3 PUFAs may also result from the elevation of *n*-3 PUFA-derived NAEs (DHEA, *N*-eicosapentaenoylethanolamine (EPEA)), which has been observed in several tissues and blood [52,56,57], as well as of the corresponding monoacylglycerols [58] and other monoacylamides [59], which possess anti-inflammatory and anticancer actions and potential cardiometabolic- and neuroprotective effects independent of cannabinoid receptors [60,61,62,63]. A recent study comparing DHA and EPA supplementation in diet-induced obese mice and type 2 diabetic patients found significantly increased levels of DHEA and EPEA in both circulation and adipose tissue, but decreases in AEA and 2-AG were only observed in mice [64]. Of note, this study by Rossmeisl et al. utilized *n*-3 PUFAs as triglycerides; however, when provided mostly as phospholipids (from krill powder) to obese men, circulating AEA levels were reduced along with triglycerides [53].

Gut microbes (collectively termed the microbiome (µB)), are not a group of commensalist microorganisms living within animals but, rather, many are mutualists, benefiting the host in a variety of ways such as aiding in energy harvesting and digestion, modulating the immune system, and influencing many aspects of metabolic health, including weight, adiposity, and lipid and glucose metabolism [65]. The µB responds quickly to dietary interventions [66], and westernized diets are linked to dysbiosis (an imbalance of microbial communities) and associated with obesity, which is generally characterized by decreased bacterial diversity with an increase in the Firmicutes/Bacteroidetes phyla ratio [67,68]. Alterations in the µB are associated with other aspects of metabolic syndrome, including dyslipidemia, hypertension, and insulin resistance (reviewed extensively in [69,70,71,72]), and their consideration for the development of targeted therapies for “precision health” plans has recently been suggested for diabetes [73,74]. Like the eCBome, the gut µB is modified by dietary fatty acids, including supplementation with *n*-3 fatty acids from fish oil and krill oil [22,75,76]. Although few studies have assessed the effects of α-linolenic acid, at least one clinical trial has indicated that α-linolenic acid-rich oils can modify the µB at the genera level [77].

In the study by Everard et al. discussed above, the chronic high-fat-diet-induced changes in the jejunum eCBome lipid levels were associated with significant alterations in the gut µB, with the proportions of 19 bacterial genera identified as being significantly modified [18]. The same group had previously shown that 4 weeks of high-fat-diet feeding increased 2-AG levels in the ileum which was also associated with an altered µB [78]. High-fat-diet-induced µB changes were associated with increased CB1 expression in the colon whereas FAAH was increased in the jejunum [79]. Thus, it appears that µB alterations in response to high-fat diets impacts upon the intestinal eCBome directly which, under obesity-inducing conditions, increases gut barrier permeability, subsequently resulting in increased circulating bacterially derived lipopolysaccharide (LPS) that subsequently modulates adipose tissue eCBome and functionality (reviewed in [80]).

## 4. Dietary Fiber and Prebiotics: Improving Gut Barrier Function through the Endocannabinoidome

The health benefits of dietary fiber have been extensively studied and reviewed, and there is little doubt that higher fiber is beneficial for cardiovascular disease, supporting prevalent recommendations that fiber intake be increased in order to maintain a healthy diet [81]. The positive effects of fiber on obesity and metabolic syndrome are believed to be intimately linked to alterations of the gut µB [23,82]. Increasing attention is being paid to “prebiotic” fiber, which is non-digestible by the host but is metabolized by gut microbiota, resulting in an alteration of the composition and/or activity of the µB, producing bioactive metabolites (such as short-chain fatty acids) that provide physiological benefits to the host [83].

One of the main positive effects of prebiotics is in regulating intestinal epithelial barrier permeability, in which short-chain fatty acids play a crucial role. The term “leaky gut” has been used to describe the phenomenon in which the tight junctions within the intestinal epithelial lining are compromised, leading to the movement of bacterially derived LPS into circulation, resulting in metabolic endotoxemia-induced inflammation that is associated with obesity [84,85]. Supplementing the diets of genetically (*ob*/*ob*) or diet-induced obese mice with the prebiotic oligofructose increases *Bifidobacterium* species and *Akkermansia muciniphila* in association with improved gut barrier function and decreased inflammation [79,86,87]. Similarly, women with type 2 diabetes who were given oligofructose-enriched inulin (10 g/day) for 8 weeks had significantly lower circulating levels of LPS and other inflammatory markers, along with decreased fasting glucose and glycosylated hemoglobin [88]. Finally, administration of pasteurized *A. muciniphila* improved insulin sensitivity and reduced total plasma cholesterol levels [89].

The eCBome has been found to regulate intestinal permeability. Using the same genetic model discussed above (*ob*/*ob* mice), Muccioli et al. showed that CB1 antagonism partially rescued tight junction integrity within the intestinal epithelium and reduced plasma LPS levels, while CB1 agonism in wild type mice increased gut permeability [79]. Further, blocking CB1 activity in mice on an obesity-inducing diet not only inhibited the development of obesity and improved glucose homeostasis, as expected, but also decreased intestinal permeability as evidenced by reduced circulating LPS levels in association with decreased adipose tissue inflammation and circulating inflammatory cytokine profile, indicating a decrease in systemic inflammation [21]. Importantly, these changes were observed in conjunction with an increase in the relative amounts of intestinal *A. muciniphila* and decreased *Lachnospiraceae*. The reduction in metabolic endotoxemia induced in *ob*/*ob* mice fed oligofructose correlated with decreased colonic CB1 expression and AEA levels, with the latter presumably due to increased expression of the AEA catabolic enzyme FAAH [79]. Thus, CB1 regulation of gut permeability, under the influence of the µB, is another mechanism by which CB1 regulates inflammation in addition to direct proinflammatory effects such as, for example, the stimulation of proinflammatory cytokine release from macrophages, which has developmental consequences for type 2 diabetes [90,91]. These results collectively support the notion that the cardiometabolic health effects of dietary prebiotic fiber is associated with alteration of the gut microbiota and intestinal eCBome, resulting in decreased intestinal permeability and the ensuing metabolic endotoxemia/systemic inflammation.

## 5. TRPV1: Linking the Endocannabinoidome to the Metabolic Benefits Attributed to Spicy Food

The consumption of spicy food has been associated with overall decreased mortality and significant reduction in hazard ratios for deaths caused by ischemic heart diseases and, in the case of the consumption of fresh chili peppers, reduced diabetes [92]. Capsaicin is the active component endowing chili peppers with their spiciness, due to activation of transient receptor potential vanilloid-1 (TRPV1) cation channels. TRPV1 channels primarily respond to noxious heat (>42 °C), but are also modulated by several eCBome members (including long-chain-saturated NAEs, monoacylglycerols, *N*-acyldopamines, and *N*-acyltaurines) [33]. Several human studies have indicated the various metabolic benefits of dietary capsaicin, which improved postprandial glucose handling in both healthy individuals and overweight individuals and women with gestational diabetes [93,94,95]. While a meta-analysis of capsaicin studies supported the positive effects of this dietary component on energy expenditure and appetite regulation, the overall effects were very small and more evident at high doses [96]. In rodent models, oral capsaicin is able to combat diet-induced obesity, insulin resistance, and hepatosteatosis [97]. The positive metabolic effects of capsaicin appear to be mediated by both TRPV1 and PPARα [97,98]. However, the role of TRPV1 in obesity and associated side effects—especially dysregulation of glucose homeostasis—is complex, as indicated by contrasting results from *Trpv1*^−/−^ mice in diet-induced obesity, in which both beneficial [99] and detrimental [100] effects have been observed. These differences may be due to variations in the diets used between studies or the ages of the mice, as *Trpv1*^−/−^ mice have been shown to have increased activity at young ages, but decreased activity at older ages, in association with increased weight gain [100,101].

Capsaicin and TRP channels have also been linked to the gut µB. The antiobesity effects of capsaicin have been associated with changes in the gut µB, including also increases in *A. muciniphila* [20,102,103]. The gut µB appears to have a causative role in mediating capsaicin antiobesity effects as gut µB transplantation from capsaicin-treated to germ-free mice replicated the capsaicin-dependent antimetabolic endotoxemia effects, which were mitigated by antibiotics in capsaicin-treated mice [104]. These changes were defined by decreases in lipopolysaccharide (LPS)-producing, gram-negative bacteria and LPS biosynthetic genes, and increases in short-chain fatty acid (SCFA)-producing bacteria, such as *Lachnospiraceae*, *Ruminococcaceae*, and *Roseburia*, as well as decreased colonic CB1 expression [104]. Accordingly, TRPV1 has been suggested to counteract increased intestinal permeability in vitro [105]. Most interestingly, in a human study, different µB enterotypes (different gut µB ecosystems) of participants were associated with the extent of capsaicin-mediated positive metabolic effects. Capsaicin increased the Firmicutes/Bacteroidetes ratio and *Faecalibacterium* abundance more prevalently in participants with the *Bacteroides* enterotype than the *Prevotella* enterotype, in combination with increased serum incretin (GIP and GLP-1) levels, which stimulate insulin production, and decreased LBP, which was assessed as a marker of inflammation [106]. As in the case of eCBs and CB1 receptors, also the communication between TRPV1 and the gut µB seems to be bi-directional. In fact, the visceral antinociceptive effects of the probiotic *Lactobacillus reuteri* has been attributed to inhibition of TRPV1 activity in mesenteric neurons [107], indicating also that the eCBome may play a significant role in mediating the activity of microbial influences on the gut–brain axis, at least with respect to pain.

## 6. Sunlight Effects on the Endocannabinoidome: A Role for Vitamin D?

Vitamin D deficiency represents a global health issue, with over a billion people being deficient [108], largely due to inadequate sun exposure. Yet, significant levels of deficiency still occur in populations living in areas of abundant sunlight [109]. Vitamin D is found only in a few foods and is thus a common dietary supplement recommended by health authorities, especially in winter months [110]. Several aspects of the metabolic syndrome are associated with vitamin D deficiency, including obesity, dyslipidemia, insulin resistance, hepatosteatosis, and hypertension [111]. The causal role of vitamin D in the pathophysiology of these aspects of the metabolic syndrome is not known, but the gut µB also appears to play a significant role. In a mouse model of diet-induced obesity, vitamin D deficiency aggravated high-fat-diet-induced insulin resistance and hepatosteatosis along with inflammation. These results occurred in conjunction with mucosal breakdown within the ileum, endotoxemia and dysbiosis with increased levels of pathogenic *Helicobacter hepaticus*, and decreased levels of the metabolically beneficial *A. muciniphila* [112]. Vitamin D receptor knockout mice also develop dysbiosis, exemplified by an alteration in the ratio of Bacteroidetes/Firmicutes phyla with increases in Lactobacillaceae and Lachnospiraceae families [113]. However, while the mechanisms remain to be determined, UVR has recently been found to alter the mouse gut µB independently of vitamin D [114].

Endogenous vitamin D is produced upon UV irradiation of 7-hehydrocholesterol in skin, which is then further metabolized, mostly in the liver and kidney, to produce bioactive 1,25-dihydroxyvitamin D_3_ [115]. The skin contains not only 2-AG and AEA, but also several other NAEs in both the dermis and epidermis [116]. Whether the skin provides a significant source of circulating eCBome mediators remains to be determined. However, in vitro exposure of melanocytes to low doses of UVB upregulates CB1 mRNA expression and increases the levels of AEA, PEA, and 2-AG in keratinocytes [117]. Further, 6 weeks of cutaneous UV exposure increased circulating 2-AG levels in both light- and dark-skinned people, without significantly altering NAEs [118]. This finding was in apparent contrast to results obtained earlier by Magina et al., who found that in psoriasis patients, whole-body narrowband UVB therapy resulted in a decrease in AEA plasma levels without affecting 2-AG [119]. The differences in these results may have been due to a variety of factors, including the fact that the employed UV radiation regimens differed between the studies and that Madina et al. studied effects only in psoriasis patients.

Vitamin D deficiency in mice increased pain sensitization and decreased CB1, but increased CB2 and PPARα in the spinal cord along with increased AEA and DHEA [120]. In the colon, 2-AG was significantly decreased together with microbial diversity, leading to an increased Firmicutes/Bacteroidetes ratio and lower levels of *A. muciniphila*. Treatment of vitamin D-deficient mice with the PPARα agonist and AEA congener, PEA, reversed the observed pain sensitization in conjunction with an increase in the levels of several microbial genera, including *A. muciniphila* [120].

Taken together, these studies suggest that sunlight exposure, and the elevation in vitamin D levels that results from it, modify the eCBome as well as the µB. Whether there is a link between the two remains to be determined. For this reason, and given that these alterations are associated with µB changes that are believed to impact on metabolic health, such as increased Firmicutes/Bacteroidetes ratios and the presence of low *A. muciniphila* levels, it will be interesting to investigate if µB–eCBome crosstalk plays a significant role in regulating obesity and associated metabolic complications downstream of vitamin D.

## 7. Effects of Exercise on the Endocannabinoidome

Exercise is the second pillar, together with the diet, which maintains metabolic health. Viscerally obese men who underwent a lifestyle modification program that included the addition of regular exercise for one year had significant improvements in several metabolic parameters as well as reduced circulating 2-AG and, to a lesser extent, AEA levels [121]. These latter alterations were very likely associated with decreased adiposity. However, while physically active men have higher lymphocyte FAAH activity than sedentary controls, suggesting higher eCBome tone within these cells, basal circulating levels of AEA, PEA, and 2-AG were not found to be different from those of sedentary males [122]. By contrast, in a study of normoweight and obese women whose activity was tracked over 6 days, while 2-AG was associated with BMI, as expected, AEA and OEA levels were positively associated with moderate–vigorous physical activity [123].

In contrast to the scarcity of data on the effects of chronic physical activity on basal eCBome mediator levels, much more research has been conducted on their response to acute exercise. Many studies have shown that physical activity quickly increases circulating AEA, but not 2-AG, levels in humans ([124,125,126] and reviewed in [127]). However, a recent study found that 2-AG, and not AEA, increased after exercise [128], and this discrepancy with previous studies may be due to the fact that the participants fasted before exercising. Interestingly, AEA increases only appear in response to medium-intensity exercise [124]. Heyman et al. showed that similar to AEA, PEA and OEA also increase during and after exercise and, in fact, are more responsive to lower intensity exercise than AEA [126]. The source of these eCBome mediators remains to be determined. However, in rats, exercise alters the levels of many NAE metabolic enzymes within the adipose tissue [129]. It has been suggested that AEA and related NAEs exert positive metabolic effects in muscle, such as improving glucose uptake and mitochondrial activity by acting at the PPARγ and TRPV1 eCBome receptors [127]. Further, exercise may modulate AEA levels directly in muscle, as has been found in the extensor digitorum longus muscles of rats [129].

Several physiological mechanisms by which exercise affects mood have been proposed, including increasing endorphins, altered mitochondrial function, and thermogenesis, as well as modulation of the endocannabinoid system [130]. The notion that increased AEA levels may be, in part, responsible for feelings of euphoria associated with exercise is supported by the finding in mice that exercise increased AEA and OEA, but not 2-AG or PEA, levels in association with decreased GABAergic neuron CB1-dependent anxiety [131]. In fact, exercise also increases eCB tone in the brain. Mice with free access to a running wheel for 8 days had increased AEA levels and CB1 binding site density in the hippocampus [132]. Furthermore, wheel running in mice results in potentiated CB1 activity within the striatum, playing a protective role against stress [133], which does not appear to be simply due to increased CB1 expression, as chronic exercise does not alter the levels of this receptor in any part of the brain [134]. Similarly, a recent study showed that singing increased circulating AEA, PEA, and OEA levels in association with improved positive mood [135]. In the same study, the effects of 30 min of cycling were also examined, and significantly increased OEA levels were observed, while AEA and PEA only showed trends towards increases. The lack of statistical increases in AEA, commonly observed in other studies, may have been due to the intensity of the cycling or the relatively small sample size. Further, exercise addicts, which have increased negative mood in response to exercise deprivation, also have lower basal circulating AEA levels than non-addicted regular runners, and exercise withdrawal and reintroduction only decreases and increases AEA levels, respectively, in non-addicts [136]. The lack of response of AEA in exercise addicts suggests that perhaps their increased amount of exercise is a homeostatic attempt to increase eCB tone.

Recent evidence indicates that exercise and the µB interact with each other (reviewed in [24]). Germ-free mice have decreased exercise performance as compared to conventional controls, and reintroduction of a single bacterial species (*Bacteroides fragilis*) partially reversed this [137]. While the sample sizes were small, Petriz et al. found that moderate exercise differentially changes the µB in wild type Wistar, obese Zucker, and spontaneously hypertensive rats, suggesting that exercise-induced changes in the µB may be dependent on the metabolic state of the host organism [138]. Similarly, high-intensity interval training of high-fat-diet-fed mice altered the µB differentially along the gastrointestinal tract with the most significant changes found in the distal regions [139]. Interestingly, exercise reversed the high-fat-diet-induced decrease in microbial diversity and the Bacteroidetes/Firmicutes ratio, which are indicative of obesity [139]. Furthermore, fecal microbiota transplant from exercised mice to mice on a high-fat diet resulted in improved metabolic parameters, suggesting that that µB can confer, at least in part, the benefits of exercise [140]. However, a more recent study found that high- or medium-intensity training had no effect on the µB of obese Zucker rats [141]. In humans, studies on professional rugby players found that their µBs were more diverse than sedentary controls and produced more short-chain fatty acids, though these changes were also associated with dietary differences [142,143]. However, other studies have found that independent of diet or BMI, higher levels of cardiorespiratory fitness correlated with higher µB diversity and short-chain fatty acid production [144]. Similarly, independent of diet, six weeks of endurance exercise in overweight women significantly altered the µB of participants with an increase in *A. muciniphila* [145], which has been shown to increase the levels of eCBome monoacylglycerols, including 2-AG [87] and, as mentioned above, to be regulated by both CB1 and TRPV1 activity [21,103]. To date, no studies have examined the potential link between the eCBome, exercise, and the gut µB. However, given that activities of several eCBome receptors (CB1, TRPV1, PPARα) have been linked to µB changes [21,103,146], it is possible that their modulation through exercise-induced changes in eCBome mediator levels may play a role in exercise-induced changes in the µB, or vice versa.

## 8. Cannabis Use and Metabolic Health

The principal psychoactive component of marijuana/cannabis (THC), one of the most commonly used recreational drugs the world over, acts mainly through CB1 activation (reviewed in [25]). Given the strong association of CB1 and its ligands AEA and 2-AG with several aspects of metabolic syndrome and obesity in general [10], it is somewhat counterintuitive that cannabis use is generally associated with an improved metabolic phenotype. Analysis of the NHANES survey from 2005–2010 found that current and past cannabis use is generally associated with significantly lower odds of metabolic syndrome [147]. Combined examination of two large epidemiological studies (NESARC and NCS-R) concluded that chronic cannabis users had significantly decreased adjusted prevalence rates of obesity, from 22%–25% in non-users to 14%–17% in users [148]. A very recent prospective analysis of NESARC data supports the above, finding that cannabis use is inversely associated with BMI increases over 3 years [149]. Several large studies have also shown inverse associations between cannabis and diabetes [150,151,152], which were corroborated by a Swedish study involving 18,000 participants, though the observed protective effects on diabetes were attenuated when adjusted for age [153]. Interestingly, these associations, observed in large heterogeneous populations, are also observed in Inuit from the Canadian north, a relatively isolated ethnic group in which the decreased weight associated with cannabis use was found to account for an association with improved glucose metabolism [154]. Cannabis use is also associated with reduced prevalence of both alcoholic and non-alcoholic fatty liver disease [155,156]. It should be noted that cannabis use is not always associated with positive metabolic outcomes; among individuals with type 1 diabetes, it is correlated with an increased risk in ketoacidosis [157], subclinical atherosclerosis (but only among cigarette smokers) [158], and mortality in patients with myocardial infarction, despite having lower rates of diabetes and hyperlipidemia [159].

The positive metabolic effects of cannabis have been attributed to the downregulation of CB1 in response to chronic cannabis use/THC exposure. Post-mortem analysis of chronic cannabis users’ brains found decreased CB1 (*CNR1*) mRNA and ligand-binding in several brain regions [160], and in vivo positron emission tomography (PET) imaging similarly showed globally decreased CB1 availability compared to controls [161]. Chronic THC administration to rats decreases 2-AG and AEA levels in the striatum, but increases AEA levels in the limbic forebrain [162], and in chronic cannabis users, AEA levels are decreased in cerebrospinal fluid while 2-AG levels are increased in the serum as compared to infrequent users [163]. The mechanism by which eCB levels were altered in these studies are unknown; however, ex vivo treatment of placental explants with THC for long (72 h) but not short (24 h) periods of time increased AEA levels, concomitant with a counterintuitive decrease in NAPE-PLD levels and a trend for increased FAAH levels [164]. In hepatocytes, THC increased both AEA and 2-AG levels, presumably by blocking the activity of fatty acid binding protein 1 (FABP1), which can act as an eCB “chaperone”, allowing eCB enzymatic degradation [165]. While THC does not appear to be able to inhibit eCBome catabolic or anabolic enzymes, several other phytocannabinoids do, though at relatively high concentrations [166], suggesting that their combination within cannabis may contribute to its ability to alter eCBome mediator levels.

In agreement with epidemiological studies, chronic THC administration to mice inhibited the development of obesity in response to a high-fat diet [167]. However, other cannabis-produced phytocannabinoids are also able to elicit positive metabolic effects. Delta-9-tetrahydrocannabivarin (THCV) is a CB1 antagonist and cannabidiol (CBD) is a CB1 negative allosteric modulator,) and both are TRPV1 agonists as well as acting on other receptors (reviewed in [25]). THCV markedly improved glucose metabolism in genetically, and diet-induced obese mice [168] and as did CBD in a genetic model of type 1 diabetes [169] Similarly, a clinical study found decreased fasting glucose levels in participants treated twice per day with 5 mg of THCV [170], whereas both THCV and CBD reduced hepatic triglyceride content in genetically obese mice [168,171].

More than 25% of non-antibiotic drugs induce dysbiosis of the µB [172]. There is limited evidence that cannabis use can modulate the gut µB. To date, only two studies have investigated the effects of cannabis use on the human gut µB. Panee et al. assayed the stools of 19 lifetime cannabis users and 20 non-users for the relative abundance of only two specific genera, *Prevotella* and *Bacteroides*, given that they are main determinants of human enterotypes [173]. They found that non-users had an average 13-fold higher *Prevotella*/*Bacteroides* ratio than cannabis users, which has been associated with plant-based as compared to animal-based diets [174]. This raised the possibility that the observed changes were attributed to alterations in the diets of users vs non-users, consistent with observations that cannabis users consume fewer fruits and more animal products and have higher caloric intake but, paradoxically, have similar nutrient serum status and lower BMIs than non-users [175]. In a second study, archived anal swabs were used to assess the µBs of HIV-positive individuals [176]. Cannabis use in these individuals was also associated with alterations in bacterial populations, including a decreased abundance of *Prevotella* as well as *Acidaminococcus* and *Dorea*, the latter two of which have been associated with obesity [177,178], along with increased abundances of other genera. The role of the *Prevotella* genus in metabolic health is complicated, due likely to the genetic diversity between individual species and, thus, several conflicting studies exist on its association with obesity, diabetes, and NAFLD, while others report positive correlations with improvements in various metabolic parameters (recently reviewed in [179]).

Chronic treatment with THC reduced weight and fat mass gain as well as energy intake in diet-induced obese but not lean mice, in association with alterations in the gut µB, which included increased levels of *A. muciniphila* and inhibition of the obesity-induced shift in the Firmicutes/Bacteroidetes ratio [167]. In an experimental autoimmune encephalomyelitis mouse model meant to mimic multiple sclerosis, a combination of the phytocannabinoids THC and CBD attenuated the induced inflammation and disease scores and significantly modulated the µB, decreasing the levels of *A*. *muciniphila* [180]. Fecal material transplantation confirmed that the protective effects were mediated by changes in the µB. Further unpublished data suggest that THC-mediated effects on the µB may be due, in part, to alterations in the host immune system, which has a complex interaction with the µB throughout a host’s lifespan [181].

It is still unclear if the modulation of the µB by THC is dependent on CB1 activity. However, inhibition of CB1 with the inverse agonist rimonabant alters the µB composition of diet-induced obese mice, including increasing *A. muciniphila* in conjunction with metabolic parameter improvements [21]. Further, an adipose tissue-specific knockout of a major NAE anabolic enzyme, NAPE-PLD, which reduced local OEA, PEA and SEA, but not AEA levels, inhibited adipose tissue browning, and led to increased weight gain, glucose intolerance, and dyslipidemia in addition to exacerbating diet-induced obesity [19]. These effects were associated with an alteration in the gut µB which when transferred to germ-free mice, partially reproduced the phenotype, and are therefore likely to be CB1-independent, as the affected NAEs are ligands for other eCBome receptors, including TRPV1, which, as indicated above, impacts upon metabolic health, at least in part through alteration of the µB. Taken together, the above studies indicate that cannabis use—through its psychoactive constituent, THC, and non-psychoactive phytocannabinoids—potentially impacts upon metabolic health, in part by modulating µB constituents.

## 9. Conclusions

In summary, we have reviewed several examples of how the lifestyle–eCBome–μB triangle, with its multifaceted aspects, is likely to play a fundamental role in both metabolic health and metabolic syndrome (Figure 2). It is likely that several healthy and “bad” lifestyle habits, in synergy with other environmental factors, independently affect both eCBome signaling and the μB, and hence help in determining their correct or defective control of energy metabolism, respectively. It is also possible, however, that eCBome and μB crosstalk—which has not yet been fully explored—directs the manner in which lifestyle cues result in virtuous or vicious circles that can respectively counteract or accelerate the development of metabolic syndrome. The molecular aspects of the lifestyle–eCBome–μB triangle, therefore, need now to be fully elucidated in order to exploit this knowledge for new lifestyle (e.g., nutritional, physical activity, etc.) and pharmacological interventions aimed at combating the appearance of one or more of the metabolic syndrome features that together contribute to the development of type 2 diabetes and cardiovascular risk factors.

## Figures and Tables

**Figure 1 nutrients-11-01956-f001:**
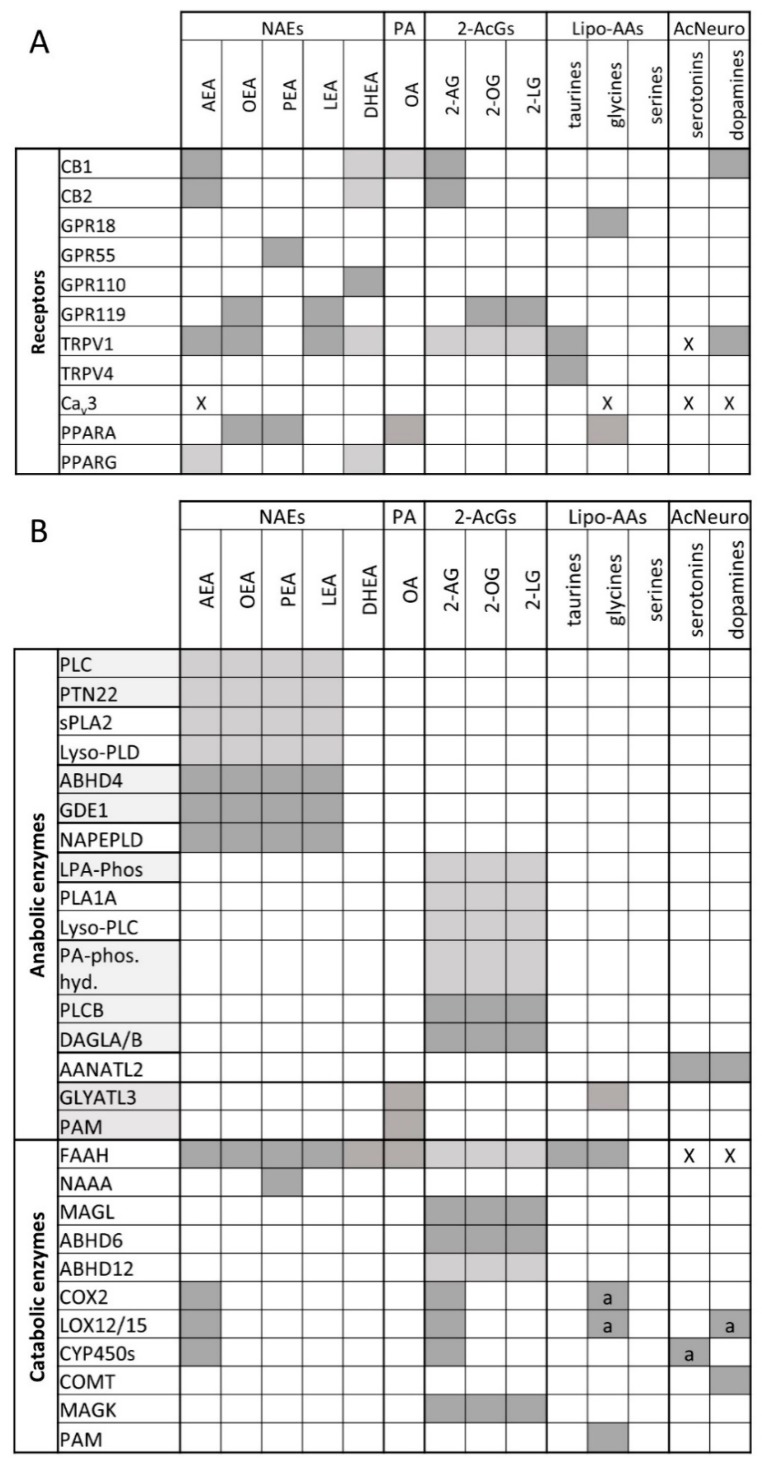
Endocannabinoidome mediators and their receptors (**A**) and anabolic and catabolic enzymes (**B**). Interactions are indicated by dark shaded boxes, and anabolic enzymes that function in concert are grouped together; “X” indicates inhibitory interactions; “a” indicates that enzymes only function with arachidonoyl homologs. A lighter shade of gray indicates a lower interaction with the receptors or a lesser role of the enzymes in biosynthesis or degradation.

**Figure 2 nutrients-11-01956-f002:**
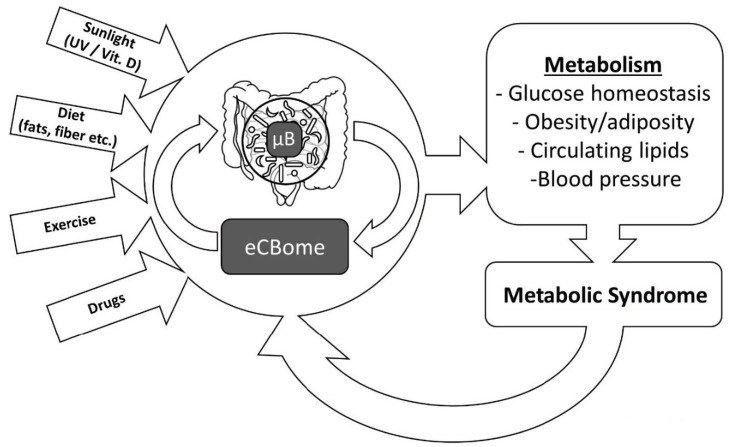
The endocannabinoidome–microbiome axis as a mechanism through which lifestyle choices affect various aspects of metabolism, which in turn may lead to the metabolic syndrome when dysregulated. This can, in turn, impact on both endocannabinoidome and microbiome-mediated signaling and, ultimately, also on lifestyle, thus creating potential vicious circles.

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
