# Peer review of "Lifestyle and Metabolic Syndrome: Contribution of the Endocannabinoidome"

_nutrients, 2019, doi:10.3390/nu11081956_

Round 1

Reviewer 1 Report

The authors did an excellent job in providing a comprehensive review of a topical issue, the interactions between the endocannabinoidome and environmental factors, including the gut microbiome, contributing to the metabolic syndrome. The review is well written and informative, and it provides a lot of food for thought beyond simply summarizing what has been published in this burgeoning field.

The following comments should be considered:

Lines 90-99: The description of the AEA biosynthetic machinery is incomplete without briefly mentioning the existence of alternative pathways with a key reference for each (Simon & Cravatt, JBC 281:2645, 2006; Liu et al., PNAS 103:1345, 2006). This is all the more warranted, as the authors do show the relevant components of these pathways in Fig. 1B, under anabolic enzymes and also in view of the well documented lack of a major reduction in tissue AEA levels in NAPE-PLD ko mice. Lines 257-291: In highlighting the novel and interesting findings of Muccioli et al. on CB1 regulation of gut permeability and, consequently, systemic inflammation via LPS/TLR4, the reader gets the impression that this is the only mechanism by which CB1 activation is pro-inflammatory. The authors should add that CB1 activation also has direct inflammatory effects (i.e. independent of LPS), as exemplified by the M1 polarization of infiltrating macrophages and the resulting release of proinflammatory cytokines, which has been demonstrated both in in vivo and in vitro paradigms (Jourdan et al., Nat Medicine 19:1132, 2013). Increased ROS production in the vascular endothelium is another mechanism by which CB1 activation contributes to a pro-inflammatory milieu without the necessary involvement of LPS (Rajesh et al. Br J Pharmacol 160:688, 2010). Lines 392-398: the possible relationship between circulating EC levels and exercise-related mood changes is purely correlational and thus represents weak evidence, it also ignores other possible players raised in the literature, such as beta-endorphin

Minor:

line 184: in a strict sense, AA is not a precursor but a constituent of 2-AG (unless the authors believe that 2-AG can be generated by condensation of AA and glycerol). Line 201: DHEA should be spelled out the first time it is used, to avoid confusion with dihydroepiandrosterone, also commonly referred to as DHEA. Line 182: ‘..a imbalance the..’ should be ‘an imbalance in the..’ Line 24: ‘…believe to…’ should be ‘…believed to…’ Line 363-366; this sentence makes no sense, should be rewritten. Line 465: ‘…able to illicit positive….’ Should be ‘…able to elicit positive…’. At a number of places, the letter μ is missing from the acronym μB

Extending parts of discussion including adding a few references, and minor text editing

Author Response

We thank the reviewer for their valuable insights.  We have incorporated all their suggestion as outlined below

Lines 90-99.  We have expressed more clearly the existence of alternate pathways in endocannabinoid metabolism and have listed the very relevant examples given by the reviewer in line 99: “While the enzymes mentioned above are considered to be the canonical ones that regulate endocannabinoid levels, it must be noted that other pathways have also been identified (see Figure 1B, recently reviewed in [1]).  For example, AEA may be synthesized by the combined action of ABDH4 and GDE1 [2] and PTPN22 [3].”

Lines 257-291. We have made clear the CB1 regulation of intestinal permeability is not the only mechanism by which it regulates inflammation, and provided an example.  Line 300: “Thus, CB1 regulation of gut permeability, under the influence of the µB is another mechanism by which CB1 regulates inflammation in addition to direct pro-inflammatory effects, such as for example the stimulation of proinflammatory cytokine release macrophages, which has developmental concequences on type 2 diabetes [4,5].”

Lines 392-398:  The reviewer is quite right, therefore we have added the following at line 390 “Several physiological mechanisms by which exercise affects mood have been proposed, including increasing endorphins, altered mitochondrial function and thermogenesis as well as modulation of the endocannabinoid system [1].”

Minor:

Line 184 (now 188) has been amended as suggested

OEA, PEA, LEA, DHEA and EPEA have now all been defined the first time they are used in the manuscript

Line 182 (now 186) has been corrected as suggested

Line 242 (now 245) has been corrected as suggested

Line 363 to 366 (now 370) has been changed to: “However, while physically active men have higher lymphocyte FAAH activity than sedentary controls, suggesting higher eCBome tone within these cells, basal circulating levels of AEA, PEA and 2-AG were not found to be different from sedentary males.”

Line 465 (now 477) has been corrected as suggested

Re: µB- I can’t find places where it is missin. I wonder if it was a problem with rendering of the document sent to the reviewer. Indeed other Greek letters don’t seem to render in the file I downloaded from the website.

Reviewer 2 Report

Review for 563434-manuscript, by Vincenzo Di Marzo and Cristoforo Silvestri submitted to Nutrients.

Metabolic syndrome is increasingly common in western populations. It is a group of conditions that occur together and involve high levels of cholesterol, triglycerides, and blood sugar, as well as excess of body fat around the waist and increased blood pressure. It is now well accepted that the broader endocannabinoid system with the polyunsaturated fatty acids, their derivatives and metabolic products, as well as the related proteins, plays a major role in the body’s metabolism and energy balance.

The present review summarizes examples as of how diet, sun light, physical exercise, and the use of cannabis, can modulate the development of metabolic syndrome by modifying the cross-talk between the broader endocannabinoid system (or “endocannabinoidome”) and the gut microbiome. The manuscript is written with care, I really enjoyed reading it, and to my opinion will capture the interest of a wide audience, including scientists (such as clinicians, pharmacologists, and medicinal chemists) and non-scientists as well.

Author Response

We thank the reviewer for his/her kind comments and are very happy that they enjoyed reviewing the manuscript.